# Burnout Syndrome among Otorhinolaryngologists during the COVID-19 Pandemic

**DOI:** 10.3390/medicina58081089

**Published:** 2022-08-12

**Authors:** Nora Šiupšinskienė, Brigita Spiridonovienė, Agnė Pašvenskaitė, Justinas Vaitkus, Saulius Vaitkus

**Affiliations:** 1Department of Otolaryngology, Academy of Medicine, Lithuanian University of Health Sciences, LT 44307 Kaunas, Lithuania; 2Faculty of Health Sciences, Klaipeda University, LT 92294 Klaipeda, Lithuania

**Keywords:** burnout syndrome, otorhinolaryngologist, depersonalization, Maslach Burnout Inventory (MBI)

## Abstract

*Background and Objectives*: To determine the prevalence of burnout syndrome among otorhinolaryngologists in Lithuania and investigate associations with sociodemographic and professional factors during the COVID-19 pandemic. *Materials and Methods*: Burnout was measured using the validated Lithuanian version of the Maslach Burnout Inventory. Demographic characteristics and professional characteristics were collected utilizing an anonymous questionnaire. *Results:* Eighty otorhinolaryngologists (ORL group) and 30 information technology professionals (the control group) were enrolled in this study. A high level of professional burnout in at least one of the subscales was observed in 82.5% of the ORL group subjects. Depersonalization and burnout syndrome were more frequently detected with increasing age in the ORL group (r = 0.2, *p* < 0.04). Greater satisfaction with salary and working environment resulted in a lower burnout incidence (r = 0.31, *p* = 0.001). *Conclusions*: During the COVID-19 pandemic, the incidence of burnout syndrome has been high among Lithuanian otorhinolaryngologists. Demographic and professional characteristics are significantly related to burnout syndrome among Lithuanian otorhinolaryngologists.

## 1. Introduction

The coronavirus disease discovered in 2019 (COVID-19) in China remains a major health system crisis [1]. Otorhinolaryngologists are specialists that have direct contact with patients infected by severe acute respiratory syndrome coronavirus 2 (SARS-CoV-2) and have a high risk of becoming infected, as they perform mucosal or aerosol-generated procedures daily (flexible/rigid endoscopy, sample taking, tracheostomy procedures, surgeries) [2]. Moreover, changed working conditions involving the usage of special equipment (FFP3/N95 masks, disposable and fluid-resistant gloves, gowns, glasses, and full-face shields) and an increased workload have created new challenges. Working in such a high-risk environment poses dangers not only to physical health but also to mental well-being (in terms of increased symptoms of stress, depression, and anxiety) which should be considered.

Burnout is a syndrome involving emotional exhaustion, depersonalization, and a diminished sense of personal accomplishment, which is primarily determined by stress at work [3]. In 1981, Christina Maslach introduced the Maslach Burnout Inventory (MBI), a tool for measuring burnout syndrome which is the most widely used means of burnout syndrome assessment to date [4]. Maslach defined burnout syndrome as emotional exhaustion resulting from stress caused by interpersonal interaction [5]. The model proposed by Maslach encompasses three dimensions (subscales) of burnout: emotional exhaustion, depersonalization, and low personal accomplishment at work [6]. This condition is included in the 10th revision of the International Classification of Diseases (ICD-10): the term ‘burnout’ is described as a “Burnout-state of total exhaustion” [7].

It is essential to investigate professional burnout among healthcare workers during outbreaks to prevent any immediate or long-lasting implications. For this reason, specialists who are directly involved with the COVID-19 pandemic—otorhinolaryngologists—were selected for the present study. According to the literature, in the USA, the prevalence of burnout among otorhinolaryngologists during the COVID-19 pandemic was 21.8% [8]. However, the prevalence of moderate-to-high burnout among academic otorhinolaryngologists in the USA can range from 70% to 75%, demonstrating that academic otorhinolaryngologists might be especially vulnerable [9]. In addition, some studies that have investigated associations between burnout and the presence of physical diseases have found strong links between exhaustion and depersonalization as well as musculoskeletal, cardiovascular, and other physical diseases [10]. Moreover, burnout in healthcare workers is associated with poorer self-rated health, increased depression, increased anxiety, sleep disturbances, and impaired memory [11].

To date, there have been no data available concerning the mental health of Lithuanian otolaryngologists. Only a few studies regarding the mental health of other medical specialists have been reported, indicating highly increased burnout rates as well as experience of severe stress and low job satisfaction [12]. In the context of other European countries, not enough research on professional burnout syndrome among otorhinolaryngologists has been presented.

This study aimed to determine the prevalence and associated risk factors for burnout syndrome among otorhinolaryngologists in Lithuania during the COVID-19 pandemic and to compare the effect of the pandemic on other specialties.

## 2. Materials and Methods

### 2.1. Ethics Statement

The study protocol was approved by the Kaunas Regional Bioethics Committee for Biomedical Research, Lithuanian University of Health Sciences (LUHS) (approval no.: BE 2-2, 17 May 2020). All study procedures were carried out in accordance with the Declaration of Helsinki. An informed consent form was obtained from all participants.

### 2.2. Design

This cross-sectional, survey-based, national study was conducted from July to December of 2020. The study involved randomly selected otorhinolaryngologists and IT professionals from the five biggest cities of Lithuania working at national hospitals (Lithuanian University of Health Sciences, Kauno Klinikos, Vilnius University Hospital Santaros Klinikos, National Hospital of Vilnius, University Hospital of Klaipeda, National Hospital of Klaipeda, National Siauliai Hospital, Panevezys National Hospital).

### 2.3. Sample Calculation Size

According to data reported by the Lithuanian Otorhinolaryngologists Society (lit. Lietuvos otorinolaringologų draugija), there are 266 otorhinolaryngologists in Lithuania (http://otorinolaringologai.org/home/istorija/otorinolaringologija-lietuvoje/, accessed on 31 May 2022). For this study, we randomly selected every third Lithuanian otorhinolaryngologist who attended scientific conferences organized by the Lithuanian Otorhinolaryngologists Society of Lithuania. Every third randomly selected IT professional (from 87 in total working in selected hospitals) was involved in this study.

The sample size calculation was based on the frequency, with a 5% probability of error and 95% reliability, and was calculated according to the formula for sample size calculation in cross-sectional studies [13]. It was calculated that the collected sample size was sufficient to reach a statistical power of 80% or higher. Furthermore, within a cross-sectional study, a sample size of at least 60 participants is recommended [14].

### 2.4. Study Population

There were 110 subjects involved in the study: 80 otorhinolaryngologists (92.5% otorhinolaryngologists and 7.5% otorhinolaryngology residents) composing the ORL group and 30 informative technology (IT) professionals. The criteria for choosing IT professionals for inclusion in the control group for this study were as follows: (1) the IT professional’s work is not recognized as involving direct contact with clients (remote-working conditions); (2) the IT professional does not have direct contact with COVID-19 patients; (3) the IT professional usually works individually and their work does not require working as part of a team; (4) working hours do not include night shifts.

For this study, IT professionals working in information technology departments in selected hospitals (no working-from-home conditions) were enrolled.

Both study groups were required to present their sociodemographic data. Salary and satisfaction with work environment were measured on the visual analog scale (VAS) from 0 to 5 (0—completely dissatisfied, 1—dissatisfied, 2—moderately satisfied, 3—satisfied, 4—sufficiently satisfied, 5—fully satisfied). Additional questions were applied for the ORL group regarding type of hospital, practice setting, academic status, patients per week, surgeries per week, and whether the individual in question worked in the private or public sector.

### 2.5. Burnout Measure

The phrase “professional burnout” was interpreted using the Lithuanian version of the MBI. This tool was chosen as it is considered to be the most commonly used implement in similar studies [15]. The MBI instrument has been already translated to Lithuanian, validated, and utilized in professional burnout studies in Lithuania [12,16,17]. The Lithuanian version of MBI is available to purchase together with the original MBI license [12].

The Lithuanian 22-item MBI version is also divided into three subscales: the 9-item emotional exhaustion (EE) scale, the 5-item depersonalization (DP) scale, and the 8-item lack of personal accomplishment (PA) scale. For this study, instead of a classic 7-point Likert type scale, a 5-point Likert type scale was chosen, as it has been most recommended by the researchers for reducing the frustration levels of respondents and increasing response rate and response quality [18]. Moreover, it has been suggested that a 5-point scale is more appropriate for European surveys [19]; a 5-point Likert scale was utilized as an effective approach to investigate the assessment of burnout syndrome influencing factors among doctors providing medical care in Lithuania [20]. The study groups were asked to answer each item on a self-completed Likert-type scale type with 5 points (1—strongly disagree, 2—disagree, 3—neither agree nor disagree, 4—agree, 5—strongly agree).

According to the literature, there is no comprehensible agreement on how to interpret burnout based on the MBI normative scores. Grunfeld et al. and Wisetborisut et al. determined burnout as high scores in any subscale (EE, DP, or PA) [21,22]. Moreover, Ramirez et al. and Tironi et al. defined burnout in terms of high scores in all three subscales [23,24]. Furthermore, according to recent studies, burnout is defined as a high score in either the EE or DP subscales and a low score in the area of PA [25].

For the present study, higher scores in the EE and DP subscales and lower scores in the PA subscale indicated a higher burnout symptom burden. The adjusted normative scores for the MBI subscales are presented in Table 1.

### 2.6. Statistical Analysis

Statistical analysis was performed using the SPSS/W 22.0 software (Statistical Package for the Social Sciences for Windows, Inc., Chicago, IL, USA). The Student’s *t*-test was used for testing hypotheses about the equality of means. For testing hypotheses about independence, the chi-square test was performed. To assess the correlations between variables, Pearson’s and Spearman’s correlation coefficients (r) were applied. The differences among means were evaluated by Analysis of Variance (ANOVA). The findings were considered statistically significant when *p* < 0.05.

## 3. Results

### 3.1. Demographics

Table 2 presents sociodemographic data for the ORL and control groups. Study groups were homogeneous according to age and gender (*p* = 0.063 and *p* = 0.082, respectively).

The working year experience in the ORL group was statistically significantly higher compared to the control group (*p* < 0.001). However, the control group subjects were statistically significantly working more hours per week compared to the ORL group (*p* = 0.005). Despite that, all control group subjects were fully satisfied with the working environment; meanwhile, 93.5% of the ORL group subjects were not satisfied with the working environment (*p* < 0.001). Moreover, satisfaction with salary was statistically significantly higher in the control group—80.0% of the control group subjects were sufficiently satisfied, whereas in the ORL group just 12.5% of subjects were sufficiently satisfied with salary (*p* < 0.001). The members of both groups made almost no reference to emotionally traumatic events in the last 6 months (divorce, loss of a relative, job loss, etc.) (*p* = 0.553) (Table 2).

Considering that the practice setting can impact the results of the ORL group (not all otorhinolaryngologists perform surgeries or undertake night shifts), we selected otorhinolaryngologists who had different daily occupations: otorhinolaryngologists working only in outpatient offices (43.8%), otorhinolaryngologists working only in departments of otorhinolaryngology (15.0%), and those who work in both settings (outpatient office and department of otorhinolaryngology) (41.2%). We would like to emphasize that otorhinolaryngologists working only in outpatient offices did not provide surgical treatment but received more patients (70.4 per/week), while otorhinolaryngologists working in both settings had 59.7 (33.1) patients per week. However, these results did not differ statistically significantly (*p* = 0.148). The mean number of surgeries performed by otorhinolaryngologists who worked in departments of otorhinolaryngology was 3.2 (4.1) per week. In addition, to obtain certain results in the present study, we involved otorhinolaryngologists who work in different types of hospitals, public (university, national, and regional hospitals) and private. Among otorhinolaryngologists working in university hospitals, 8.8% also worked in academic areas (Table 2).

### 3.2. MBI Score

The most significant results for the emotional exhaustion (EE) subscale: 35.0% of the ORL group expressed that they agreed with the statement about feeling emotionally drained from work, while just 3.3% of the control group agreed with the statement (*p* < 0.001); a quarter of the ORL group subjects (25.0%) agreed that they got up already fatigued, while no subjects in the control group agreed with this statement (*p* < 0.001); only 3.3% of the control group agreed that working with people all day was a strain; however, in the ORL group, 26.3% of the subjects agreed that working with people all day was a strain for them (*p* < 0.001), and 11.3% of the ORL group subjects agreed with the statement that working directly with people put too much stress on them, while no subjects in the control group agreed with this statement (*p* = 0.05) (Appendix A).

The most significant results for the depersonalisation scale (DP) subscale: 23.8% of the ORL group subjects and 50.0% of the control group subjects strongly disagreed that they felt like they treat some patients/clients as if they were impersonal subjects (*p* < 0.001); in the ORL group, 11.3% of the subjects agreed, and 3.7% of the subjects strongly agreed that they had become more callous toward people since they took their job, whereas no subjects in the control group agreed or strongly agreed with this statement (*p* < 0.001); in the ORL group, 61.3% of the subjects agreed that they felt patients blamed them for some of their problems, whereas one-fifth (20.0%) of the control group subjects also agreed that they felt clients blame them for some of their problems (*p* < 0.001) (Appendix A).

The most significant results for the personal achievement subscale: almost half (46.7%) of the control group subjects strongly agreed that they dealt very effectively with the problems of their clients, whereas just 7.5% of the ORL group subjects strongly agreed with the statement (*p* < 0.001); one-third (33.3%) of the control group strongly agreed that they felt themselves to be positively influencing other people’s lives through their work; however, just 3.7% of the ORL subjects strongly agreed with the statement (*p* < 0.001); almost half (46.7%) of the control group subjects strongly agreed that they felt very energetic, whereas only 7.5% of the ORL group subjects strongly agreed that they felt very energetic (*p* < 0.001); 60.0% of the control group subjects strongly agreed that they felt exhilarated after working closely with their clients; however, only 5.0% of the ORL subjects strongly agreed with this statement (*p* < 0.001); 40.0% of the control group subjects strongly agreed, and 46.7% agreed that they had accomplished many worthwhile things in their job; however, only 6.2% of the ORL group strongly agreed with the statement (*p* < 0.001); 60.0% of the control group subjects strongly agreed that in their work they dealt with emotional problems very calmly, whereas only 3.7% of the ORL group subjects strongly agreed with the statement (*p* < 0.001) (Appendix A).

### 3.3. Professional Burnout

Using Maslach’s three subscales of burnout, 41.3%, 20.0%, and 71.3% of the ORL group subjects were found to have experienced a high incidence of professional burnout according to the EE, DP, and PA subscales, respectively. In addition, moderate professional burnout was observed in 55.0%, 68.8%, and 26.3% of ORL group subjects via EE, DP, and PA subscales, respectively (Table 3).

### 3.4. Correlation between MBI Subscale Scores and Sociodemographic Characteristics

Depersonalization and a high incidence of burnout syndrome were more frequently detected with increasing age in the ORL group (r = 0.2, *p* < 0.04). In addition, greater satisfaction with salary resulted in lower burnout incidence (r = 0.31, *p* = 0.001). Additionally, greater satisfaction with work environment resulted in a lower burnout rate (r = 0.32, *p* = 0.001). No significant associations between the response rates concerning other factors were determined. No significant correlations were identified in the control group.

## 4. Discussion

This study is the first study that has addressed burnout among Lithuanian otorhinolaryngologists during the COVID-19 pandemic. Moreover, no studies have previously been performed on the prevalence of burnout syndrome among Lithuanian otorhinolaryngologists. The findings of this study support the concern that otorhinolaryngologists are experiencing a high level of burnout—the prevalence of burnout was 51.3%. Furthermore, in terms of the three subscales for the MBI, more than one-third (41.3%) reported high emotional exhaustion, one-fifth (20.0%) reported high depersonalization, and almost three-quarters (71.3%) reported experience of highly reduced personal accomplishment. In addition, 82.5% of the ORL group subjects were classified as experiencing a high level of burnout in at least one of the subscales. These data are essential, since otorhinolaryngologists are the specialists that have direct involvement with COVID-19 and the COVID-19 pandemic remains a global health system crisis with SARS-CoV-2 mutations resulting in different variants of the virus.

The results of studies analyzing burnout among otorhinolaryngologists during the COVID-19 pandemic are alarming. Civantos et al. analyzed mental health among otorhinolaryngologists and attending physicians during the COVID-19 pandemic using a single-item Mini-Z Burnout Assessment, a 7-item Generalized Anxiety Disorder Scale, a 15-item Impact of Event Scale, and a 2-item Patient Health Questionnaire and reported that burnout was experienced by 21.8% of physicians [8]. In addition, Civantos et al. also analyzed mental health among head and neck surgeons in Brazil during the COVID-19 pandemic using the same instruments and confirmed an incidence of burnout in 14.7% of physicians [26]. Another study performed by Momin et al. surveyed the Texas Association of Otorhinolaryngology to evaluate burnout. It was concluded that 50% of participants expressed that COVID-19 and attendant changes contributed to physician burnout in their practice [27]. Larson et al. analyzed the prevalence of associations between distress and professional burnout among otorhinolaryngologists. In their study, an abbreviated 2-item version of the MBI was developed and validated. It was clarified that professional burnout was present in 49% of otorhinolaryngologists [28]. However, there are still no studies analyzing burnout using the three subscales of the 22-item MBI among otorhinolaryngologists during the COVID-19 pandemic.

Studies analyzing burnout using the 22-item MBI among otorhinolaryngologists before the pandemic also require attention. In 2019, Attopuls et al. analyzed burnout among otorhinolaryngologists in Australia and reported that 73.3% of correspondents suffered from burnout in at least one of the three MBI subscales [29]. The results of the previous study showing that 82.5% of the ORL group subjects experienced a high level of burnout in at least one of the MBI subscales are very similar. Moreover, our results are similar to those of other studies conducted in Europe. Vijendren et al. investigated professional burnout among otorhinolaryngologists in the United Kingdom (UK) and found high incidence rates of stress and psychological morbidity (56.5%) and professional burnout prevalence (28.9%) [30]. However, in their study, the authors used the General Health Questionnaire-12 (GHQ-12) and a 9-item abbreviation of the original 22-item MBI.

A study conducted by Golub et al. among United States academic otolaryngologists demonstrated that burnout was common among academic otolaryngologists [31]. In their study, they used a 22-item MBI and revealed that a high level of burnout was observed in 4% and moderate burnout in 66% of the respondents. Additionally, it was found that women experienced a statistically significantly higher level of emotional exhaustion (EE) when compared to men. In addition, associate professors were significantly more burned out than full professors. In this study, we did not find any significant differences in professional burnout between men and women, nor were doctors differentiated according to sub-specializations, and no respondents working in private and public medical institutions were distinguished.

A study conducted by Fletcher et al. analyzed factors that may have influenced professional burnout [32]. It was indicated that young age, number of hours worked per week, and length of time in practice were statistically significant predictors of burnout. Another study conducted among residents of otorhinolaryngology in 2020 reported that increase in the number of working hours was confirmed to have resulted in an upsurge of burnout, according to the MBI [33]. Our study has also established that long working hours have a significant effect on the onset of professional burnout; however, it has also revealed that older rather than younger age has a critical effect on the development of professional burnout. This may have been due to the fact that the majority of respondents were older and that only a small number of resident physicians participated in the study. Other sociodemographic factors included in this study did not have any significant effect on professional burnout. However, this study clarified that depersonalization (DP) and burnout syndrome were more frequently detected with increasing age in the ORL group (r = 0.2, *p* < 0.04).

In this study, it was revealed that many otorhinolaryngologists (71.3%) were frustrated with their personal accomplishment at work, whereas in a study conducted by Contag et al., otorhinolaryngologists experienced moderate professional burnout but indicated high levels of personal achievement (62%) [34]. These conflicting results could be explained by the facts that this study was performed during the global pandemic situation and that, in our study, not only operating otorhinolaryngologists were included. Furthermore, according to this study, it was clarified that low salaries and an unfavorable work environment had a great impact on the burnout of otorhinolaryngologists in Lithuania. Considering why this factor could be so significantly related to professional burnout, it is possible to state that the importance of salaries may be related to the economic situation in Lithuania.

This study’s results correspond to other frontline healthcare personnel burnout rate results. Anesthesiologists play the most significant role in the COVID-19 pandemic, since they participate in multiple aerosol-producing procedures, such as pre-oxygenation, mask ventilation, laryngoscopy, tracheal intubation, and extubation. Afonso et al. utilized a 22-item MBI to evaluate burnout rate and risk factors among anesthesiologists in the United States and revealed that 59.2% (2307 of 3898) of anesthesiologists were at high risk of burnout while 13.8% (539 of 3898) met criteria for burnout syndrome [35]. A study performed by Podhorodecka et al. revealed that 73% (115 of 158) of anesthesiologists suffered from burnout during the COVID-19 pandemic in Poland [36]. Moreover, Sevinc et al. demonstrated that higher levels of anxiety and burnout were identified in younger healthcare workers in Turkey [37].

In this study, we identified only one subject (3.3%) in the control group who experienced a high level of burnout (*p* < 0.001). These results are contradicted by others reported in the literature. Kumaresan et al. demonstrated that the prevalence of burnout syndrome among IT professionals who worked from home during the COVID-19 pandemic was 95%, with a predominance in females [38]. These contradictory results could be explained by the fact that, in our study, the predominant gender of IT professionals was male (53.3%). Additionally, IT professionals selected for this study continued working in their casual environments—the information technology departments in the selected hospitals—and therefore avoided the uncertainties associated with changed working conditions, hours, and restrictions on salary. Furthermore, our study involved IT professionals who were working in the same area (health information technology issues), whereas the previously mentioned study involved IT professionals working in different areas [38]. Moreover, in the previously mentioned study, there were no working hours or working conditions mentioned. Hence, there is still a lack of data for the analysis of IT professionals’ mental health during the COVID-19 pandemic. Further studies with larger samples should be provided to understand better the risk of burnout among IT professionals.

To summarize, the main factors influencing rates of burnout among otorhinolaryngologists could be as follows: (1) direct contact with patients infected by SARS-CoV-2 and a high risk of becoming infected by performing mucosal or aerosol-generated procedures daily; (2) changed working conditions involving special equipment usage (FFP3/N95 masks, disposable and fluid-resistant gloves, gowns, glasses, or full-face shields); (3) increased workload; (4) fear of infecting family members; (5) the outcome of the pandemic remaining uncertain. From our point of view, IT professionals are mainly affected by: (1) changed working conditions (wearing a facemask during working hours); (2) the outcome of the pandemic remaining uncertain. However, in each country, the causes of professional burnout vary depending on the age, economic situation, and professional prospects of the subjects.

The strength of this study was the careful selection of investigated groups (both groups were adjusted for age and sex; also, the ORL group involved different profiles of otorhinolaryngologists). To the best of our knowledge, this is the first study to analyze burnout using three subscales of the 22-item MBI among otorhinolaryngologists during the COVID-19 pandemic. In addition, this study revealed alarming results regarding the prevalence of burnout among Lithuanian otorhinolaryngologists, suggesting that these practitioners should start adapting their lifestyles and professional habits as soon as possible to recover from burnout.

## 5. Conclusions

This is the first study to measure burnout syndrome among otorhinolaryngologists in Lithuania, and it has revealed that professional burnout among Lithuanian otorhinolaryngologists during the COVID-19 pandemic has been high. This study identified that sociodemographic characteristics, such as age, working environment, and salary, are significantly related to lower prevalence of professional burnout. We believe that the presented results may contribute to lessening professional burnout among otorhinolaryngologists in Lithuania.

## Figures and Tables

**Table 1 medicina-58-01089-t001:** The adjusted normative scores for the Maslach Burnout Inventory subscales for the present study.

MBI Subscales	High Professional Burnout	Moderate Professional Burnout	Low Professional Burnout
Emotional Exhaustion (EE) (Score: 45–9)	≥25	24–15	14–9
Depersonalization (DP) (Score: 25–5)	≥15	14–10	9–5
Personal Accomplishment (PA) (Score: 45–8)	14–8	15–24	≥25
Emotional Exhaustion (EE) (Score: 45–9)	≥25	24–15	14–9

**Table 2 medicina-58-01089-t002:** Sociodemographic characteristics of the ORL and control groups.

Characteristic	Group		*p*-Value ^3^
	ORL ^1^ *n* = 80	Control *n* = 30	
**Male**, *n* (%)	28 (35.0)	16 (53.3)	0.082
**Female**, *n* (%)	52 (65.0)	14 (46.7)	
**Age, years**; mean (SD) ^2^	53.5 (15.2)	47.3 (10.3)	0.063
**Civil status,***n* (%) Married/adult children Married/school-aged children Married/no children Single Other (divorced/widowed)	45 (56.3) 17 (21.3) 3 (3.7) 14 (17.5) 1 (1.2)	17 (56.7) 6 (20.0) - 3 (10.0) 4 (13.3)	0.970 0.882 - 0.335 0.007
**Working experience**, mean (SD) ^2^	26.6 (17.7)	7.7 (4.2)	<0.001
**Working hours**, mean (SD) ^2^	50.8 (17.3)	57.3 (8.7)	0.005
**Satisfaction with work environment** Completely dissatisfied dissatisfied Moderately satisfied Satisfied Sufficiently satisfied Fully satisfied	- - 12 (15.0) 26 (32.5) 37 (46.3) 5 (6.2)	- - - - - 30 (100.0)	- - - - - <0.001
**Salary** Completely dissatisfied Dissatisfied Moderately satisfied Satisfied Sufficiently satisfied Fully satisfied	- 17 (21.3) 34 (42.5) 19 (23.7) 10 (12.5) -	- - - - 24 (80.0) 6 (20.0)	- - - - <0.001 -
Emotionally traumatic event in the last 6 months	1 (1.6)	0 (0.0)	0.488
**Type of Hospital *** **Public** - University hospital - National hospital - Regional hospital **Private hospital**	11 (13.8) 22 (27.5) 38 (47.5) 9 (11.2)		
**Practice setting *** Outpatient Office ENT Department Outpatient office and ENT Department	35 (43.8) 12 (15.0) 33 (41.2)		
**Academic status**, *n* (%)	7 (8.8)		
**Patients per week ***, mean (SD) ^2^	65.7 (41.9)		
**Surgeries for week**,** mean (SD) ^2^	3.2 (4.1)		

^1^ ORL: otorhinolaryngologists group; ^2^ SD: standard deviation; ^3^ *p*-Value: significance level *p* < 0.05. * Only ORL group; ** Only 39 otorhinolaryngologists involved (working in ENT department and outpatient office and ENT department).

**Table 3 medicina-58-01089-t003:** Distribution of burnout subscales according to adjusted normative categories for both groups.

Subscale	Mean (SD) ^2^		Median		High		Moderate		Low	
	ORL ^1^ Group	Control Group	ORL ^1^ Group	Control Group	ORL ^1^ Group	Control Group	ORL ^1^ Group	Control Group	ORL ^1^ Group	Control Group
Emotional Exhaustion (EE)	23.7 (4.9)	15.4 (2.6)	24	15	33 (41.3)	0 (0.0)	44 (55.0)	12 (40.0)	3 (3.7)	18 (60.0)
Depersonalization (DP)	12.5 (3.2)	9.8 (2.3)	13	10	16 (20.0)	1 (3.3)	55 (68.8)	4 (13.3)	9 (11.2)	25 (83.4)
Personal Accomplishment (PA)	26.1 (6.1)	35.1 (2.8)	27	36	57 (71.3)	0 (0.0)	21 (26.3)	0 (0.0)	2 (2.4)	30 (100.0)

^1^ ORL: otorhinolaryngologists group; ^2^ SD: standard deviation.

## Data Availability

All data relevant to the study are included in the article or have been uploaded as Appendix A.

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
