# Peer review of "Burnout Syndrome among Otorhinolaryngologists during the COVID-19 Pandemic"

_medicina, 2022, doi:10.3390/medicina58081089_

Round 1

Reviewer 1 Report

The study investigates levels of burnout in otorhinolaryngologists in Lithuania during the COVID-19 pandemic, compared to a control group of IT professionals in the same time-frame. This is a cross-sectional survey study. The authors use the Maslach Burnout Inventory to measure burnout and correlate burnout level with demographic and work variables. Burnout is a timely and important topic among healthcare workers currently, as the pandemic has impacted healthcare workers' wellbeing. The concept of burnout as defined by Maslach is discussed appropriately. 

Suggestions for the authors:

1. Clarify whether the focus of the study is burnout among ORLs vs other professions, predictors of burnout in ORLs (or physicians in general), burnout in ORLs during vs prior to the pandemic, or the effect of the pandemic on ORLs vs other medical specialties

2. The claim that aerosolizing procedures single out ORLs among physicians and therefore makes them the specialty in direct touch with COVID-19 is unclear. Anaesthesiologists, for example, perform more or similar levels of aerosolizing procedures, which exposes them to at the very least similar risks. Paramedics and respiratory technicians may also fall in this group. The article can benefit from discussing whether burnout levels are similar between these specialties/occupations and how this impacts the conclusions. 

3. Information technology is actually an area where quite a bit of burnout studies have been done (see citation list below):

"Salanova, Marisa; Peiró, José M; Schaufeli, Wilmar B; ",Self-efficacy specificity and burnout among information technology workers: An extension of the job demand-control model,European journal of work and organizational psychology,11,1,25-Jan,2002,Taylor & Francis

"Shih, Sheng-Pao; Jiang, James J; Klein, Gary; Wang, Eric; ","Job burnout of the information technology worker: Work exhaustion, depersonalization, and personal accomplishment",Information & Management,50,7,582-589,2013,Elsevier

"Huang, Stanley YB; Fei, Yu-Ming; Lee, Yue-Shi; ",Predicting job burnout and its antecedents: Evidence from financial information technology firms,Sustainability,13,9,4680,2021,MDPI

"Cook, Sara L Schwarz; ",Explaining burnout: A mixed method investigation of information technology workers,,,,,2006,Capella University

"Pawlowski, Suzanne; Kaganer, Evgeny; Cater III, John; ",Mapping perceptions of burnout in the information technology profession: A study using social representations theory,ICIS 2004 Proceedings,,,73,2004,

"Jung, Eunju; ",Work stress and burnout: The mediating role of mood regulation among information technology professionals,Journal of Workplace Behavioral Health,28,2,94-106,2013,Taylor & Francis

"Zaza, Sam; Riemenschneider, Cynthia; Armstrong, Deborah J; ",The drivers and effects of burnout within an information technology work context: a job demands-resources framework,Information Technology & People,,,,2021,Emerald Publishing Limited

"Shropshire, Jordan; Kadlec, Christopher; ","I’m leaving the IT field: The impact of stress, job insecurity, and burnout on IT professionals",International Journal of Information and Communication Technology Research,2,1,,2012,Citeseer

"Sethi, Vikram; Barrier, Tonya; King, Ruth C; ",An examination of the correlates of burnout in information systems professionals,Information Resources Management Journal (IRMJ),12,3,13-May,1999,IGI Global

"Mehrtak, M; Mahdavi, A; Valizadeh, S; ",Relationship between Locus of Control and Burnout among Health Information Technology Staffs at University Hospitals in Ardabil,Journal of Health,9,4,414-422,2018,Journal of Health 

3a. It would be important to justify the choice of this control group

3b. Given that burnout in the control sample in this study seems to differ from the literature, discuss potential causes and implications of this difference

4. The level of English word use and sentence structure fluctuates throughout the paper. Suggest having a native-proficiency English speaker edit the draft.

5. Depending on the chosen focus of the article, expand the discussion on factors influencing the rates of burnout among ORLs vs IT professionals, or ORLs during the pandemic vs other medical specialties/professions during the pandemic

6. Specific suggestions/questions:

- Lines 49-51: clarify - does this suggest burnout is lower in ORLs during the pandemic? Or that academic ORLs are especially vulnerable?

- Lines 81-82 and 87-88: Were ORLs randomly selected or recruited at conferences? Or both?

- Line 94:  the last sentence of the paragraph is superfluous

- Lines 78, 89, and 101: please clarify how the control group was recruited. Were they randomly selected from among those working at hospitals or were they recruited through a private company?

- Line 102: see above - why is it that IT work is not considered emotionally debilitating?

- Table 2: Although the gender and age distribution in the two groups is not significantly different at the alpha=0.05 level, it approaches significance. Also, the actual age mean (SD) is missing for the Control group. 

- Table 2: Under type of hospital, replace "Republic" with "National"

- Table 2: Replace "Working Occupation" with "Practice Setting"

- Table 2: Clarify the meaning of "Pedagogical status"

- Section 3.2 of Results: report results more succinctly. Consider reporting by subscale, not by item, possibly with highlighting high-impact items

- Consider relegating Table 3 to appendix and keep Table 4 in the text

- Line 260-262: Sentence 2 and 3 of the paragraph - clarify whether the correlations were performed with both groups or with one only.

- Line 270: distinguish "high-level burnout" from "high prevalence of burnout"

- Line 275: again, ORLs are presumably not "the" specialists that have direct touch with COVID-19

- Line 280-285: The two quoted studies seem to indicate that ORLs have a lower incidence of burnout compared to physicians in general, which would go against the argument of the study

Author Response

Dear Reviewer,

We appreciate the revision of our manuscript “Burnout Syndrome Among Otorhinolaryngologists during COVID-19 Pandemic”. We have enclosed the original manuscript marked with all the changes made during the revision (in Track Changes). We hope that the revised manuscript will be acceptable for publication in Medicina journal.

Enclosed please also find our point-by-point response to the comments raised during the revision process.

We would like to express our sincere thanks to the Reviewers who identified areas of our manuscript that needed corrections or modifications. We would like also to thank you for allowing us to resubmit a revised copy of the manuscript.

Reviewer 2 Report

We recommend to exclude the conclusion about "each country" ( should be in the Disscussion and to include one conclusion about lower level of burnout  and correlation in the conclusion.

I suggest a better explanation about the IT comparation -group

Author Response

(The authors gave the same response as above.)
